# Visual Analytics for Climate Change Detection in Meteorological Time-Series

**Milena Vuckovic** *[iD] **and Johanna Schmidt** [iD]

VRVis Zentrum für Virtual Reality und Visualisierung Forschungs-GmbH, 1220 Vienna, Austria; johanna.schmidt@vrvis.at
* Correspondence: milena.vuckovic@vrvis.at

**Abstract:** The importance of high-resolution meteorological time-series data for detection of transformative changes in the climate system is unparalleled. These data sequences allow for a comprehensive study of natural and forced evolution of warming and cooling tendencies, recognition of distinct structural changes, and periodic behaviors, among other things. Such inquiries call for applications of cutting-edge analytical tools with powerful computational capabilities. In this regard, we documented the application potential of visual analytics (VA) for climate change detection in meteorological time-series data. We focused our study on long- and short-term past-to-current meteorological data of three Central European cities (i.e., Vienna, Munich, and Zürich), delivered in different temporal intervals (i.e., monthly, hourly). Our aim was not only to identify the related transformative changes, but also to assert the degree of climate change signal that can be derived given the varying granularity of the underlying data. As such, coarse data granularity mostly offered insights on general trends and distributions, whereby a finer granularity provided insights on the frequency of occurrence, respective duration, and positioning of certain events in time. However, by harnessing the power of VA, one could easily overcome these limitations and go beyond the basic observations.

**Keywords:** climate change; meteorological time-series; global warming; visual analytics; visual computing

## 1. Introduction

### 1.1. Background

The unprecedented global increase in the frequency and magnitude of extreme weather events and related consequences (e.g., heat waves, flooding and drought, severe storms, wildfires) is being recognized as one of the most pressing environmental issues and a worldwide health and lifestyle concern. The synthesis reports published by the Intergovernmental Panel on Climate Change (IPCC) confirmed that these observed transformative changes in climate system are closely tied to the anthropogenic processes and related elevated emissions of greenhouse gases (GHG) in the Earth's atmosphere [1,2]. It is a well-documented fact that GHG such as water vapor, carbon dioxide ($CO_2$), methane ($CH_4$), nitrous oxide, which occur naturally in the atmosphere, along with the synthetic fluorinated gases, which originate from a variety of industrial processes, have the tendency to absorb, store and reradiate long-wave radiation emitted from Earth's surface back to Earth's surface [3,4]. The effect is generally known as the 'greenhouse effect' and has a significant impact on energy budget of the Earth system, resulting in global atmospheric warming and chaotic weather patterns worldwide [3]. The global character of this phenomenon is mainly driven by the fact that Earth's atmosphere intermixes globally, meaning that this phenomenon is of no geographical or spatial specificity. However, the degree to which this drives site-specific environmental issues, such as the air, water and soil pollution, alongside the occurrence of landslides, fluvial flooding, wildfires in forest landscapes, to

name a few, depends on the levels of local GHG emissions, which tend to further amplify the global effect [5].

Given these far-reaching negative impacts of GHG on climate dynamics, atmospheric composition and natural and human environment, numerous national adaptation strategies are set in motion in order to promote global transformation towards a climate-neutral, primarily a low-carbon, economy [6]. These strategies and their unique goals build upon a landmark environmental agreement reached between parties to the UNFCCC (United Nations Framework Convention on Climate Change) to combat climate change and its negative impacts, which is known as the Paris Agreement [7]. The Paris Agreement's central aim relates to combined efforts to limit the global temperature rise to 2 degrees Celsius above preindustrial levels, while preferably targeting a threshold of 1.5 degrees Celsius, which is to be achieved in this century. However, even with a strong commitment by countries to make substantial and consistent reductions in climate forcing emissions, whereby the envisioned mitigation measures are already being applied worldwide, the effects of these measures drive a rather slow response in the evolution of climate change due to inherent climate inertia and internal variability [8,9].

It is thus clear that a systematic and continuous monitoring of climate system and related developments and trends is an essential prerequisite in understanding mitigation progress achieved so far, but also in detecting the degree of natural progression of climate change and respective implications for meteorological parameters. The present manuscript and related research tackles the latter aspects. More specifically, we approach climate change investigation by means of visual analytics of diverse meteorological time-series datasets. This mainly relates to discovery of hidden insights by interactive visualization and visual data mining [10].

### 1.2. Time-Series Analysis

The importance of meteorological time-series data for climate research is unparalleled [11]. These high-resolution temporal data sequences (e.g., minute-based, hourly, daily) allow for systematic detection of nonlinear dynamics, abrupt changes in sequential distributions, events and anomaly detection in a system over time. This facilitates, among other aspects, the identification of diverging trends, periodic behaviors, discrepancies in peak values and daily amplitudes. Such insights are critical when dealing with multivariate meteorological time-series, whereby distinct structural changes in a single parameter may drive an immediate or a delayed response in other parameters. These are known as interdependencies or inter-relationships between several parameters, which are often complex, non-linear, and non-uniform.

Such intricate inquiries call for applications of progressive analytical tools and techniques that go beyond conventional methods used to describe only the basic features of the underlying data (e.g., descriptive statistics). One promising approach relates to application of visual analytics (VA)—a cutting-edge analytical system of advanced computational capabilities powered by interactive and highly responsive visual representations [12]. Essentially, VA systems deliver innovative data foraging schemes and data transformation concepts capable of supporting multidimensional, multivariate, and often-ambiguous data queries applied to multifaceted data streams. This is facilitated by the pre-built interactive dashboards composed of interlinked analytical view ensembles, each enriched with a suite of robust statistical transformative techniques. Through such visual data exploration and visual data mining, VA enables the process of identifying hidden patterns, features, and phenomena that would otherwise not be easily recognized by standard algorithmic means.

In the context of meteorological time-series analysis, VA could in principle help determine the ongoing progression of warming tendencies under the climate change, the degree of stability or instability of a system, the frequency of extreme events, or develop new predictive schemes. However, even with such valuable application potentials, the actual adoption of VA practices in climate research is still fragmented.

*1.3. Overview*

Given this background, we aim to demonstrate how a VA approach can assist in climate change detection, illustrated through a systematic analysis of meteorological time-series data. We look into several past-to-current meteorological datasets delivered in different temporal intervals (i.e., monthly, hourly). Specifically, we focus on long-term (45 years) and short-term (5 years) timespans to assert the degree of climate change signal that can be derived given the varying temporal scale of the underlying data.

The selection of this particular long-term time span was driven by the fact that, when focusing on climate investigations, the data records of at least 30 years are considered appropriate to fully capture the variability of climate conditions and their tendencies. The selection of short-term time span was in part driven by the initial findings derived from the long-term records, stressing that the last 7 years show notable and consistent warming tendencies (see Section 3.1). This was further reduced to 5 years due to the data availability stemming from selected weather stations.

## 2. Materials and Methods

Given that the majority of the global population currently resides in urban areas [13], the general well-being and quality of life of urban societies is becoming increasingly important. Led by this notion, we narrowed our research focus to the progression rate of warming trends and changes in climate dynamics in urban areas. Specifically, we selected three high-density traffic-intensive cities from the Central European region (i.e., Vienna, Munich, and Zürich) that are of the same Köppen climate classification (Cfb—temperate oceanic climate) [14]. This allowed us to have more controlled background climate conditions and respective urban influences (e.g., degree of urbanization, traffic levels, population density). In general, Cfb designation denotes cool winters and warm summers, with typically lacking dry seasons as precipitation is more evenly dispersed throughout the year [15]. The selected cities also vary in size. It should be noted that we are not aiming at comparative analysis of these cities. Rather, our additional objective is to show that, independent of their size, cities are equally affected by climate change.

*2.1. Data Sources and Study Parameters*

The high-resolution (monthly, hourly) data records originate from several international open data initiatives managed by national meteorological services, in the case of Vienna and Munich, and city authorities, in the case of Zürich [16–18]. Respective weather station names and locations are the following: *Hohe Warte* (48.2490560°, 016.3556230°) for Vienna, *München-Stadt* (48.1406109°, 011.5496953°) for Munich, *Schimmelstrasse* (47.3710000°, 008.5235000°) for Zürich.

Long-term datasets represent 45 years of monthly records (from 1975 to the end of 2020). Short-term datasets represent 5 years of hourly records (from 2016 to the end of 2020). It was noted that not all the cities offer the same set of publicly available parameters (e.g., air temperature, humidity, solar radiation, wind). Thus, the first step was to cross-reference the available data parameters and find those that are present in each dataset. This was not seen as problematic as the result of our cross-reference analysis revealed an adequate pool of relevant common parameters. In the case of long-term monthly data, we reduced the focus on those parameters that are known to influence the properties of the climate system. These are referred to as atmospheric parameters and relate, in our case, to air temperature, precipitation, and solar radiation. In the case of short-term hourly data, we focused on air temperature, relative humidity, and wind speed, as these are more representative of local microclimate conditions that are more suitable for a short-term analysis. Table 1 provides an overview of selected parameters and respective aggregation levels and units. It should be noted, however, that the wind speed data was provided in both km/h (Vienna) and m/s (Munich, Zürich) units, which was accordingly converted to m/s, but also that this data

was collected from different heights. Thus, we used the power law to estimate the wind speed at a uniform height of 10 m above the ground (at the urban boundary level) [19,20]:

$$v/v_n = (H/H_n)^{\alpha}, \tag{1}$$

here, $v$ is the wind speed at target height of 10 m (denoted by H), $v_n$ is the wind speed at height $H_n$ (height of the observation), and $\alpha$ is the friction coefficient (power-law exponent), which is equal to 0.40 for high-density urban areas. The respective observation heights ($H_n$) are 35 m for Vienna, 29 m for Munich, and 7 m for Zürich.

**Table 1.** Overview of selected parameters for long-term and short-term analysis.

| Long-Term Parameters | Description |
|---|---|
| Air temperature | Monthly average, absolute maximum, and absolute minimum of daily temperatures in 2 m height [°C] |
| Precipitation | Monthly sum of total precipitation height [mm] |
| Solar radiation | Monthly sum of sunshine duration [h] |
| **Short-Term Parameters** | **Description** |
| Air temperature | Hourly temperature records in 2 m height [°C] |
| Relative humidity | Hourly humidity records in 2 m height [%] |
| Wind speed | Mean hourly wind speed [m/s, km/h] |

### 2.2. Visual Analytics System

The VA system used for the analysis is an analytical ensemble solution developed at our institution, specifically designed for interactive visual exploration of multi- and high-dimensional time-series data, including categorical and functional data. These analytical cockpits are composed of several individual interlinked computational modules, each equipped with different built-in analytical technique for data processing. They support specific tasks such as the data quality assessment (missing values, duplicate timestamps, time gaps, and anomalies), detecting a value threshold breach, but also pattern search and predictive model generation [21]. Additionally, several user-defined time- or event-based filters and additional calculations can be made, allowing the end users to perform ad-hoc queries and analyze the underlying data and key performance metrics. Our system offers an extremely responsive design, achieved through a multi-threaded architecture, allowing instantaneous feedback and smooth interaction with the data across diverse set of analytical cockpits. This means that, once a user selects a data point or a data subset of interest in one analytical module, the corresponding data points are highlighted in another module (see, for example, Figure 1). As these analytical cockpits provide an instant comparison between large sets of data, the process of identification of general temporal behaviors, outliers, and any deviations can be done with little effort. Such inquiries namely relate to the detection of unusual distributions, peak and null values, discrepancies, and general data noise. These are essential for studying complex multivariate systems, such as the climate system.

### 2.3. Missing Data

Prior to the actual analysis, the raw data was structured to meet the data format requirements of the VA system described above. During this process, it was noted that some timestamps were missing in both long- and short-term datasets for all three cities (less than 1% of the data for each parameter). This was adjusted using a custom script for filling gaps in a temporal sequence, which was written in R programming language. These fields are left empty, so as to retain the data integrity. Table 2 provides an overview of the missing instances in all datasets.

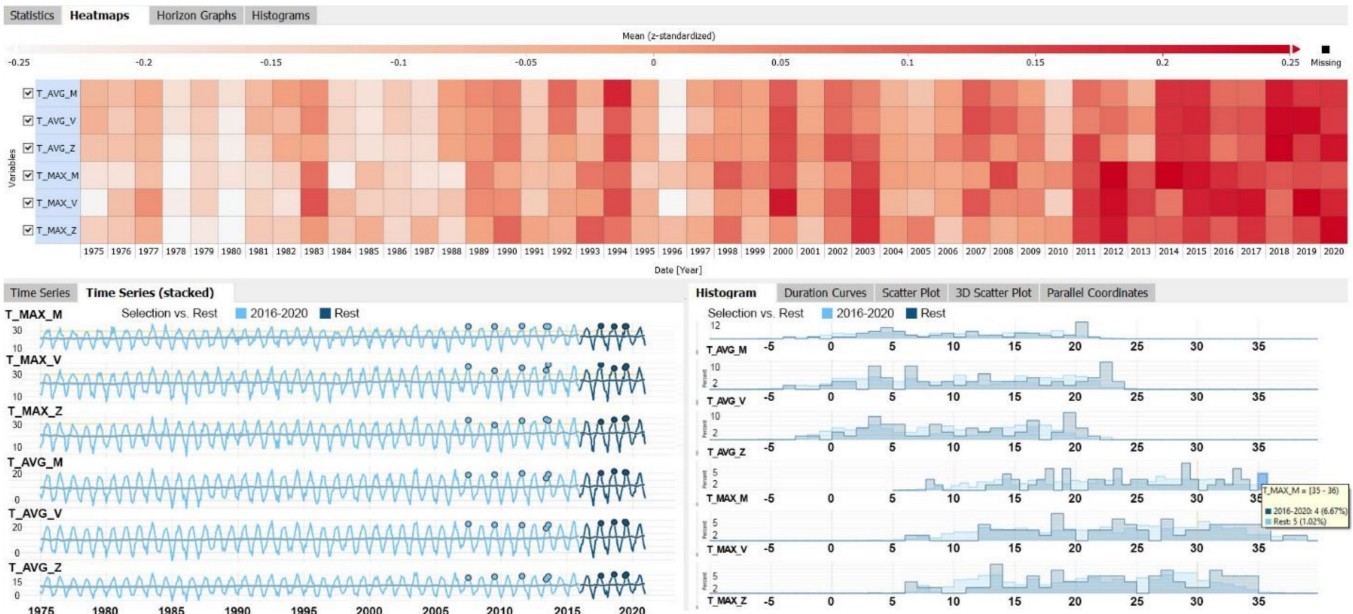

**Figure 1.** Composite view of temperature time-series (average and maximum values): heatmap view (**up**), time-series distribution (**down left**), frequency distribution (**down right**).

**Table 2.** Total number of missing timestamps (number of hours) in the data for each parameter.

| Long-Term Parameters | Vienna | Munich | Zürich |
|---|---|---|---|
| Air temperature | - | - | - |
| Precipitation | - | - | - |
| Solar radiation | - | 1 of 552 h | - |
| **Short-Term Parameters** | | | |
| Air temperature | 265 of 43,848 h | 24 of 43,848 h | 405 of 43,848 h |
| Relative humidity | 264 of 43,848 h | 27 of 43,848 h | 405 of 43,848 h |
| Wind speed | 262 of 43,848 h | 34 of 43,848 h | 390 of 43,848 h |

## 3. Results and Discussion

### 3.1. Long-Term Analysis

Going through the rather coarse level of acquired long-term data (i.e., monthly), we soon realized that by relying on such coarse granularity and conventional analysis methods (e.g., descriptive statistics, value change over time) one could only gain insights that point to the general trends and distributions. However, by using a VA approach we could effectively leverage these issues by going beyond such initial observations.

To exemplify, we investigated temporal changes using a time-based distribution enriched with a heatmap view and frequency distribution of each parameter and for each city (Figures 1 and 2). Figure 1 illustrates distribution of average and maximum temperatures. Figure 2 illustrates distribution of minimum temperature. Due to several initial observations described below, we further compared a reference timespan of last 5 years (2016–2020) to the rest of the dataset (1975–2016), in an effort to highlight the abrupt changes in the recent years when compared to the 30 years past. This is distinctly colored in shades of blue in Time-Series view in Figures 1 and 2.

Firstly, we can see a clear and consistent upward trend in temperature values present on all aggregation levels (i.e., average, minimum, maximum). This is best observed in the heatmap view, where the white-to-red color scheme illustrates the annual progression, with red denoting the highest value. In the case of the observed cities, the average temperature rose by approximately 1.5 to 2 °C, while the $T_{max}$ reached a difference of up to 5 °C, observed from the 1975 reference line. This is consistent with the current scientific

consensus stating that the Earth's climate is experiencing a rapid warming [22]. It should be noted that a part of the observed warming might indeed be attributed to climate change. However, a certain degree of this warming may be driven by the contextual urban factors, such as the degree of local urbanization and population density, traffic levels, density of urban structure, etc. We further noted that the $T_{min}$ was equally affected by the noted warming, whereby this trend was even more progressive. The respective temperature deviation (2020 vs. 1975) ranges from 5 to 13 °C between the observed cities.

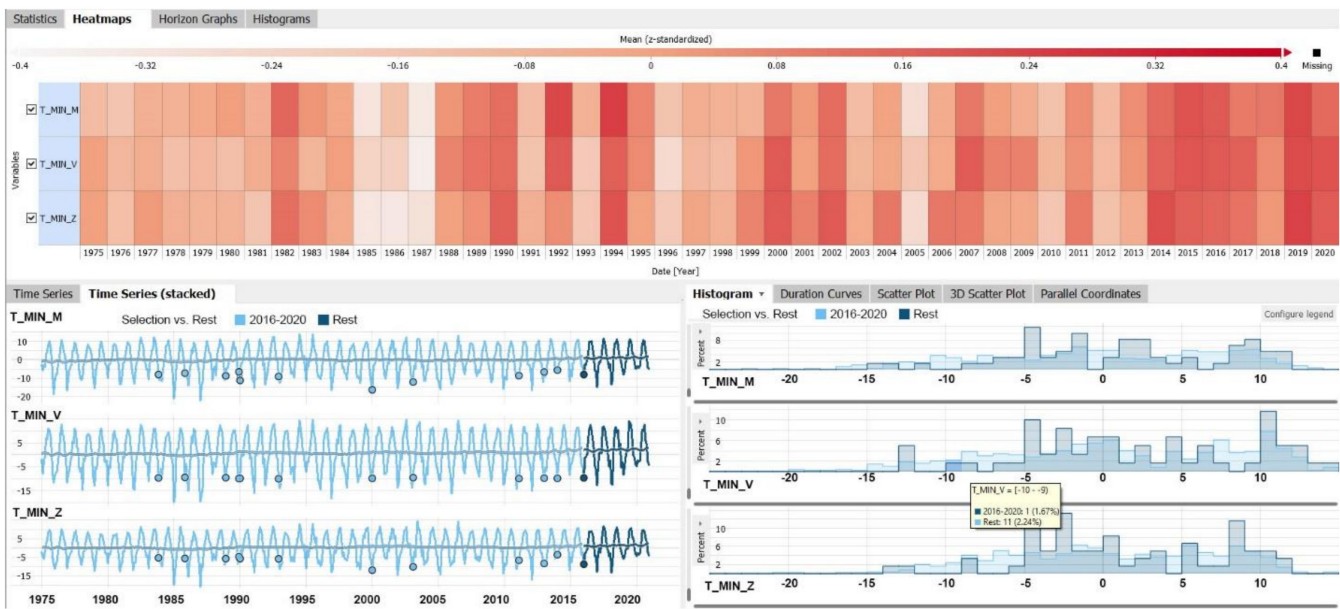

**Figure 2.** Composite view of temperature time-series (minimum values): heatmap view (**up**), time-series distribution (**down left**), frequency distribution (**down right**).

Secondly, from the frequency distribution graph (Figure 1, down right) we can see that the temperatures exceeding 35 °C are more prominent in the last 5 years and that all of such instances are clustered in the last 15 years (highlighted in the Time-Series view). Specifically, these particular insights are made possible by the inherent interactivity of the used VA system, allowing users to select data points or areas of interest in one view and get an instant visual response in another. In our case, by selecting the bars raging from 35 °C and above in the Histogram view in Figure 1, this reveals when exactly did such conditions occur in the Time-Series view. Similarly, looking at the $T_{min}$ we can see that the temperatures below −10 are rarely occurring in the last 5 years and almost not at all in the last 2 years (Figure 2, down right). Looking at the temporal scale, we can see that most of these changes happened in the last 5 to 7 years for all cities. This further indicates that we are already in a warming period that is further evolving. This can be additionally supported by a threshold breach assessment analysis carried out using a specific functionality of the deployed VA system (Figure 3). We specifically looked at the frequency distribution of those instances where a single temperature value breached the predefined lower (−5 °C) or upper (30 °C) temperature thresholds. These thresholds denote the periods of increased heat or cold stress, following the principles of a universal thermal comfort index called the universal thermal climate index (UTCI) [23]. We can clearly observe a steady progression of higher and a decline of lower thresholds breaches.

Interestingly enough, the solar radiation data revealed a steady increase of solar hours compared to the 1975 (Figure 4, above). The respective increase (2020 vs. 1975) was 316 h for Vienna, 384 h for Munich, and 578 h for Zürich. This increase was present in all meteorological seasons, being the most prominent during the periods of meteorological spring and summer. Together with the precipitation results showing a slight upward trend (Figure 4, below), the following assumption can be made: we are currently experiencing a dry period

with longer sunshine duration and more likely occurrence of sudden precipitation events (i.e., heavy rain events).

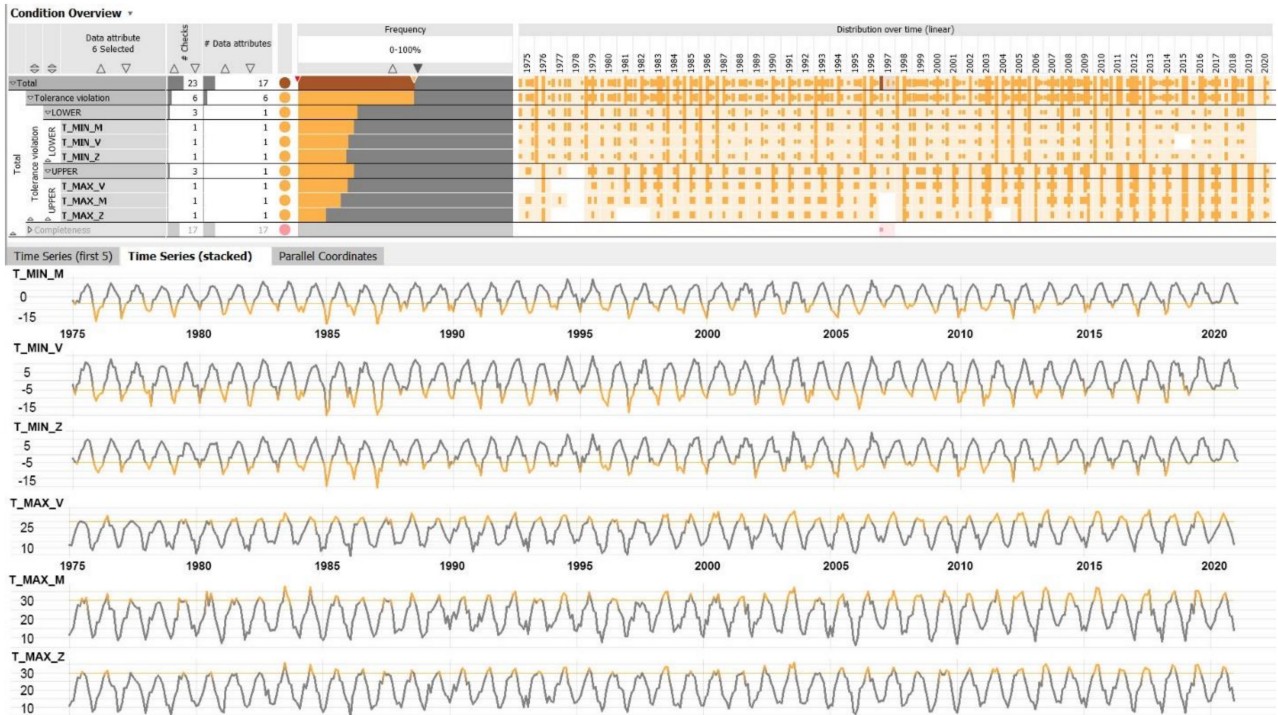

**Figure 3.** Threshold breach assessment of temperature time-series (minimum and maximum values).

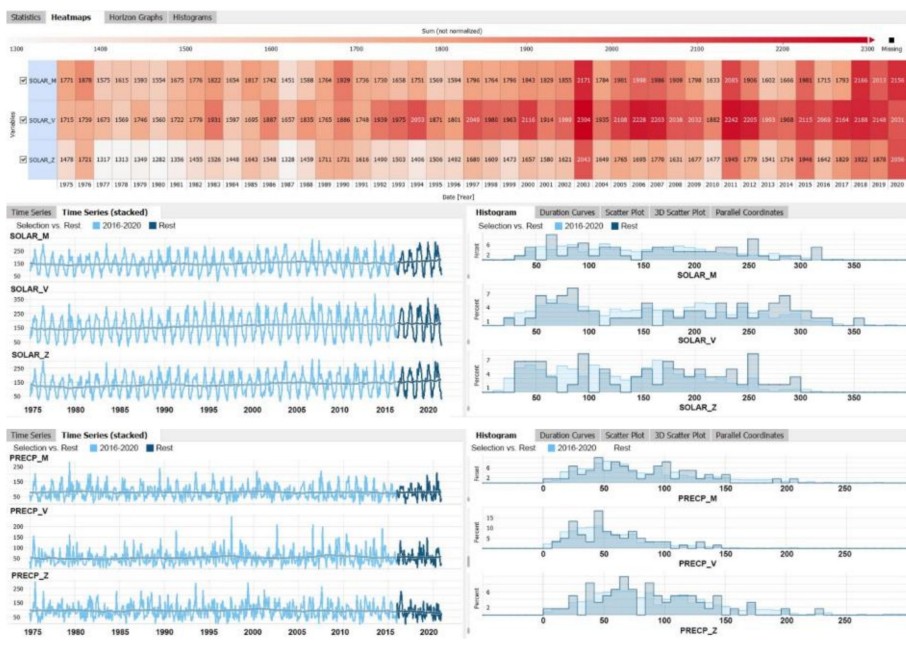

**Figure 4.** Composite view of solar (**above**) and precipitation (**below**) time-series: heatmap view (**up**), time-series distribution (**down left**), frequency distribution (**down right**).

## 3.2. Short-Term Analysis

In contrast to monthly observations, hourly records gave us several additional insights on the frequency of occurrence, respective duration, and positioning of certain events in time. We will first discuss the temperature findings, where we observed the 2020 conditions

with reference to the rest of the dataset (i.e., 2020 vs. 2016–2019). The frequency distribution of high temperature extremes in 2020 (Figure 5, right) was found to resemble the shape and the height position of the 2016–2019 curve. This further supported our previous observation regarding the evolving warming period. We thus focused more on the specifics of how $T_{min}$ was affected by such developments. Looking at Figure 5 (frequency distribution view), we can also notice that the 2020 annual temperature range appears to be notably smaller than for the 2016–2019 period. This holds true for all the cities. Namely, in 2020, $T_{min}$ does not exceed $-4\ °C$, while in the previous years the absolute $T_{min}$ reached $-10\ °C$. The records also show a higher percentage of generally higher lower temperatures ranging from zero to $10\ °C$, indicating a tendency towards generally warmer winters.

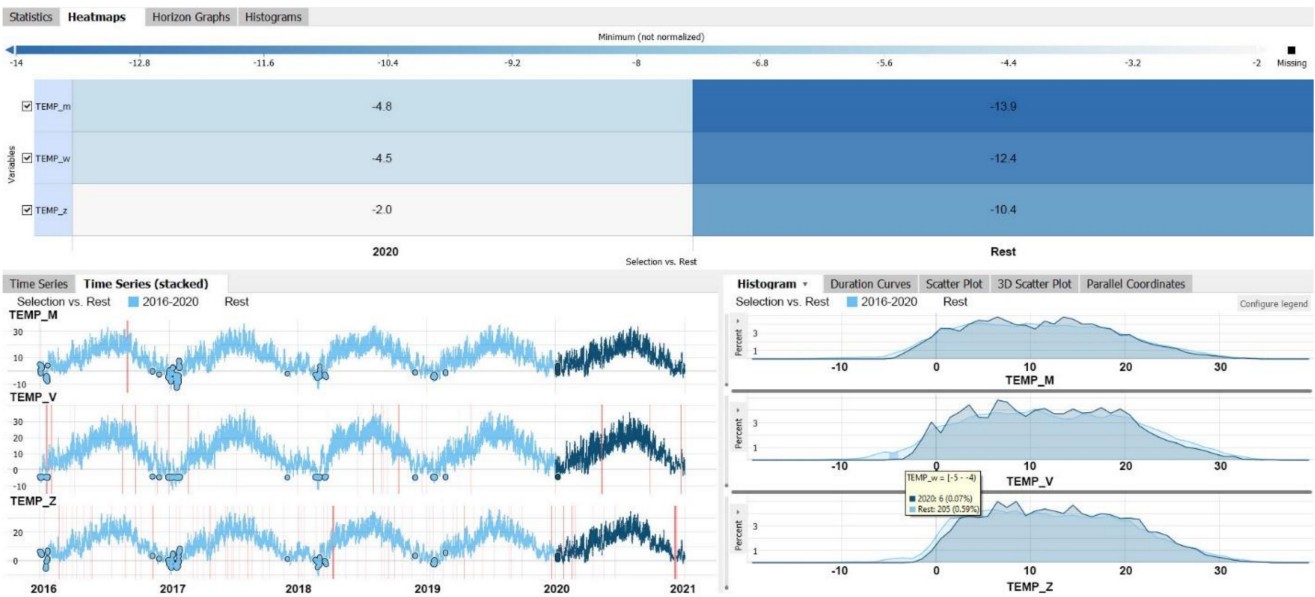

**Figure 5.** Composite view of comparative temperature time-series analysis with focus on temperature minima: heatmap view (**up**), time-series distribution (**down left**), frequency distribution (**down right**).

We further investigated when in time the properties of temperature curves started to change and how long such conditions lasted. We looked into the annual distribution of diurnal duration of temperatures breaching a predefined threshold ($30\ °C$ for $T_{max}$ and $0\ °C$ for $T_{min}$) for each city and each year (Figures 6–8). These durations are expressed in number of seconds in the day (maximum being 86,400). In these figures, the height of lines indicates the respective duration, thus longer lines denote longer duration of hot or cold conditions.

In the case of $T_{max}$ (Figures 6–8, above), we could not find a stable regularity, just several apparent trends. As such, there is an apparent trend for temperature extremes (above $30\ °C$) to be of longer diurnal duration and to start earlier in the year, observed from the 2016 base line. For example, for the month of June, an absolute diurnal $T_{max}$ duration increase (2019 vs. 2016) of around 3.5 h was noted for Zürich, 4 h for Munich, and 1.5 h for Vienna. This further insinuates the potential development of tropical nights in summer months. However, in 2020, such extremes were recorded only from the month of July on. This is namely due to the inherent irregular behavior of the climate system and a general difficulty to anticipate sudden changes due to multivariate external forcing [24].

However, in the case of $T_{min}$, the annual variations appeared to be more evident, especially in the case of Zürich and Minuch (Figures 6–8, below). We noted an apparent change in both duration and positioning of temperature minima. Specifically, the first incidences of below zero temperatures are recorded later in the year for 2020 compared to 2016, their occurrence is rarer, and their diurnal duration generally decreased.

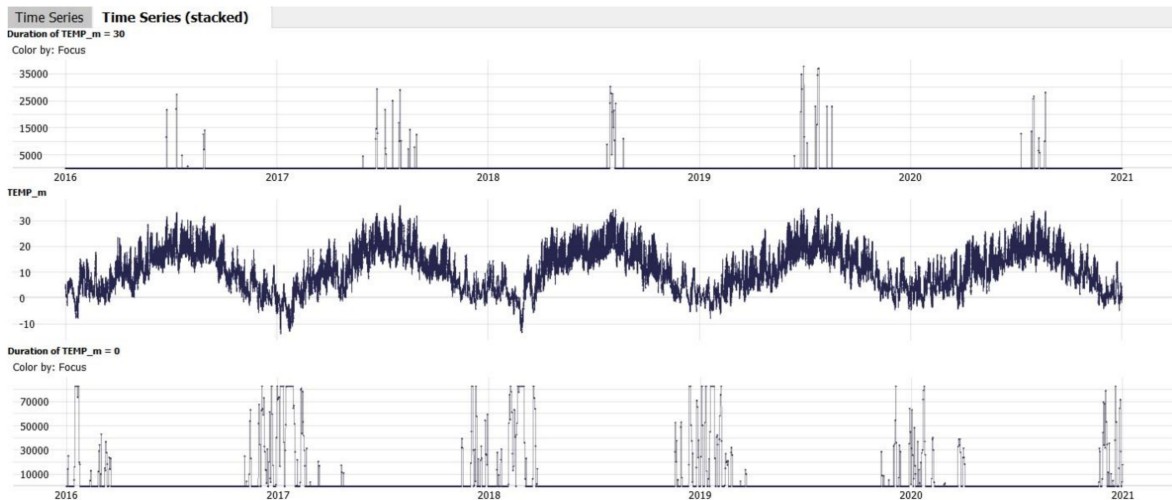

**Figure 6.** Annual distribution and diurnal duration (expressed in seconds) of temperature breaching a predefined threshold for Munich: temperature maxima (**above**), temperature minima (**below**).

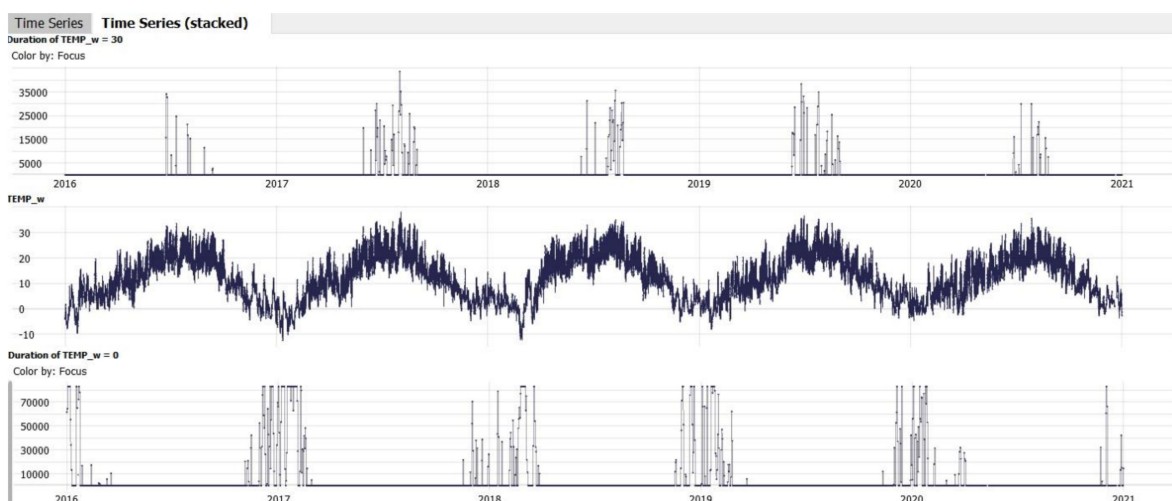

**Figure 7.** Annual distribution and diurnal duration (expressed in seconds) of temperature breaching a predefined threshold for Vienna: temperature maxima (**above**), temperature minima (**below**).

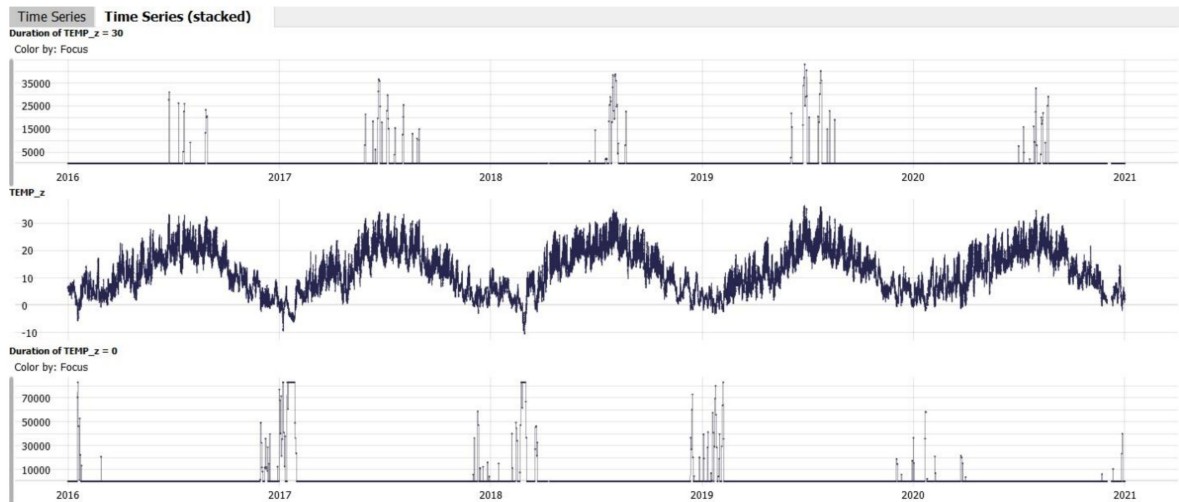

**Figure 8.** Annual distribution and diurnal duration (expressed in seconds) of temperature breaching a predefined threshold for Zürich: temperature maxima (**above**), temperature minima (**below**).

The associated multivariate pattern analysis offered a supporting study to the observed behavioral change indications in meteorological data. Here we carried out the analysis of paired instances of temperature, humidity, and wind data (Figures 9–11). For this purpose, we investigated the annual progression of hot events by a paired comparison of high temperature/low wind/low humidity events (i.e., $T_{max}$ > 30 °C, wind < 2 m/s, humidity < 50%). The low temperature/high wind/high humidity events (i.e., $T_{min}$ < 0 °C, wind > 5 m/s, humidity > 50%) were used to explore the progression of cold events. Again, we can note a rather steady development of hot events throughout the observed years (Figures 9–11, above). In some cases, such events tend to start earlier and end later in the year, suggesting thus the longer periods of heat stress. In contrast, cold events seem to be less frequent and their individual occurrence more spread out over time (Figures 9–11, below). This further points to the possible seasonal shifting or loss of seasonality altogether.

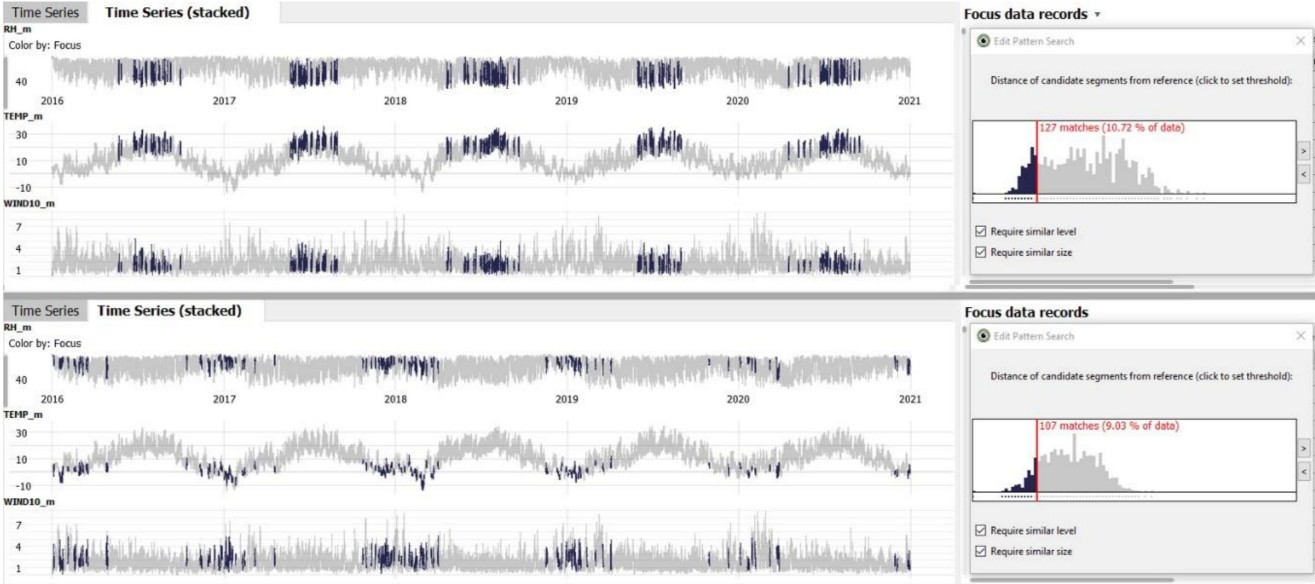

**Figure 9.** Multivariate pattern analysis of paired parameters for Munich: hot events (**above**), cold events (**below**).

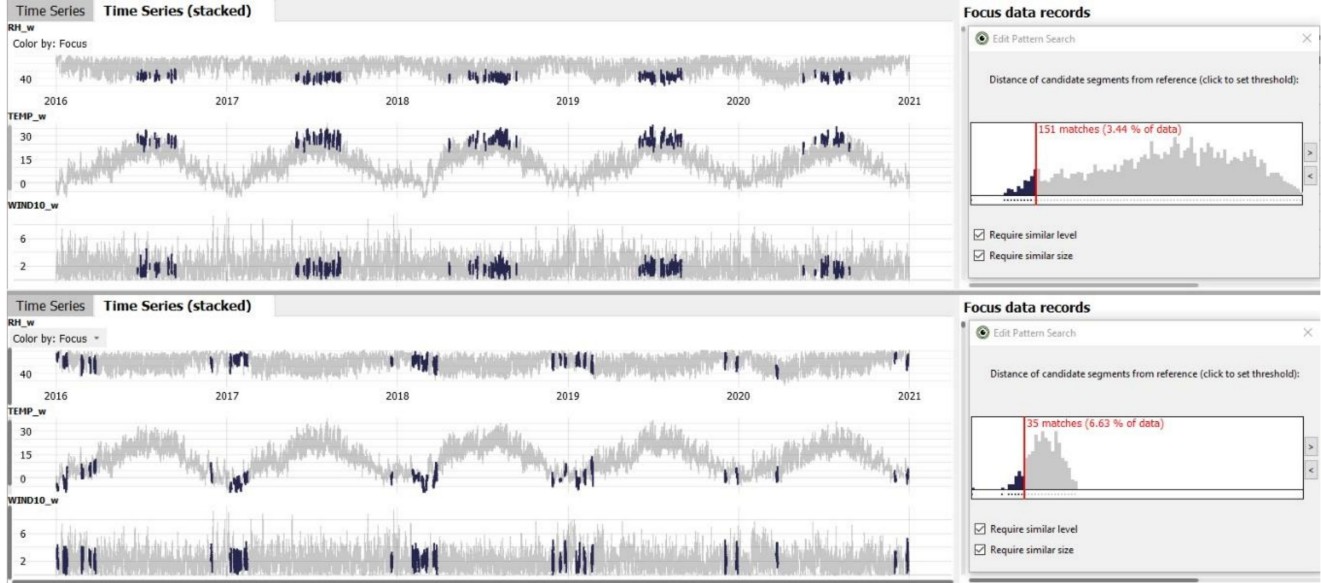

**Figure 10.** Multivariate pattern analysis of paired parameters for Vienna: hot events (**above**), cold events (**below**).

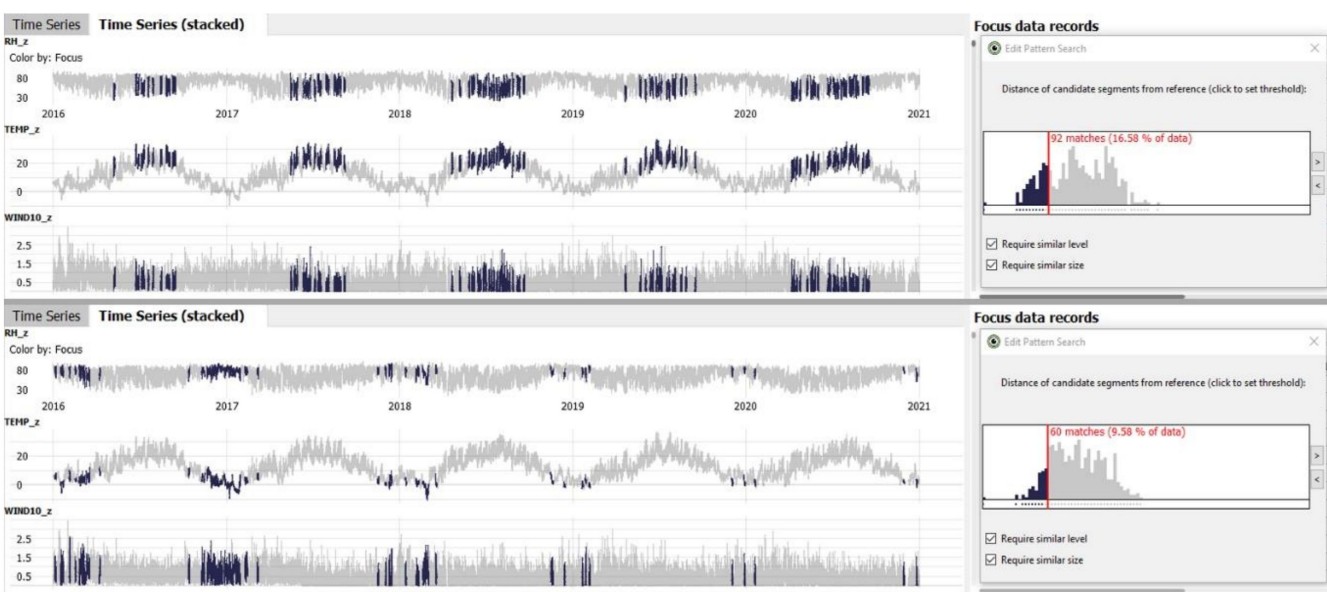

**Figure 11.** Multivariate pattern analysis of paired parameters for Zürich: hot events (**above**), cold events (**below**).

## 4. Conclusions

We carried out a systematic analysis of long- and short-term past-to-current meteorological time-series data, collected from three Central European cities, to detect insinuations of climate change in such datasets. As the underlying data were delivered in different temporal intervals (i.e., monthly, hourly), we looked at what kinds of information can be derived given such different levels of detail. To facilitate such inquiries, we used a visual analytics (VA) approach to leverage the inherent limitations of conventional descriptive statistics methods. In this, we aimed not only to highlight the overall benefits of VA in general, but also its pivotal application potential in the context of climate research. In our previous work [21], we directed attention to some promising benefits of using the VA approach in the context of meteorological time-series analysis, although with a specific focus on data structure and quality check analysis of diverse datasets, further applied on a much smaller scale (i.e., only one year worth of data). In the present contribution, we focused on how a visual data exploration may be used to detect climate change from meteorological time-series datasets of varying resolutions.

The analysis of data with coarse granularity (i.e., monthly records) initially only offered the insights on general trends and distributions. Specifically, we noted a clear warming trend from the 1975 base line, whereby both temperature maxima and minima shifted towards higher values. By further harnessing the power of VA, we could perform a comparative analysis of a specific timespan (i.e., 2016–2020) with reference to the rest of the dataset, along the distinct threshold breach assessment. This namely gave us a more detailed understanding of the nature and degree of such warming, along the frequency distribution of periods of increased heat and decreased cold stress. The majority of related high-temperature events (exceeding 35 °C) seem to be more prominent in the last 5 to 7 years for all cities. In contrast, low temperature events (below −10 °C) seem to be almost non-existent in the same reference timeframe. Solar and precipitation data further suggested a more prominent dry period with potential increased frequency of heavy rain events.

The analysis of data with fine granularity (i.e., hourly records), on the other hand, offered insights on time-dependent feature variation in data structure (i.e., duration and position of change events). We could see that the general warming trend is still evolving, with a tendency of temperature extremes (above 30 °C) to start earlier in the year and last up to 4 h longer during the day. This could also point to a likely development of tropical nights in summer months. The multivariate analysis of paired parameters (temperature, humidity,

wind) suggested a steady development of hot events and less frequent development of cold events throughout the observed years. This allows us to assume a possibility of seasonal shifting.

In conclusion, we can say that a VA approach, with its diverse semi-automated analytical capabilities, serves as a powerful tool for detecting hidden nuances in massive and multifaceted data streams, thus allowing for discovery of new relevant insights and patterns. The integration of this interactive visualization science in climate research would help scientist move forward from simple confirmatory to progressive exploratory data analysis. Here, the former commonly relies on static graphical representations of derived results, while latter pursues a continuous interaction with large and complex data for a deep understanding of the full complexity and informed reasoning about the phenomena. Such a visual data mining approach also provides the means of investigating the actual development of processes and phenomena, instead of just the derived metrics.

## 5. Future Research Directions

The application potential of visual data exploration outlined in this paper is meant to highlight the opportunities for novel approaches in climate research, which had not previously been prominent in the practice. As such, our current study aimed to use such progressive methods for detection of distinct changes in local climate depicted from raw climate data acquired from urban areas.

However, we also recognize the highly beneficial aspect of detecting which contextual urban factors (e.g., emission levels, degree of urbanization) and related mechanisms affect the changes in the local climate the most. As this requires additional sources and types of data (both numeric and categorical data), we are currently pursuing such holistic investigations through our future research efforts.

Additionally, we also aim to focus on a comparative study of meteorological records stemming from urban areas (as done in the present research) and surrounding open land (rural) areas. This latter would allow us to isolate a clear climate change signal, deprived from any anthropogenic influence on climate.

**Author Contributions:** Conceptualization, M.V. and J.S.; methodology, M.V. and J.S.; validation, M.V.; formal analysis, M.V.; investigation, M.V.; resources, M.V.; data curation, M.V.; writing—original draft preparation, M.V.; writing—review and editing, J.S.; visualization, M.V. All authors have read and agreed to the published version of the manuscript.

**Funding:** This research received no external funding.

**Institutional Review Board Statement:** Not applicable.

**Informed Consent Statement:** Not applicable.

**Data Availability Statement:** Data used in this research is a part of open data initiatives of Austria, Germany and Switzerland. Publicly archived datasets may be found at the following links: (AT) https://www.data.gv.at/katalog/dataset/wetter-seit-1955-hohe-warte-wien; http://at-wetter.tk/; (DE) https://opendata.dwd.de/; (CH) https://data.stadt-zuerich.ch/dataset/ugz_meteodaten_stundenmittelwerte.

**Acknowledgments:** VRVis is funded by BMK, BMDW, Styria, SFG (Steirische Wirtschaftsförderungsgesellschaft m.b.H. SFG) and Vienna Business Agency in the scope of COMET—Competence Centers for Excellent Technologies (879730), which is managed by FFG (Österreichische Forschungsförderungsgesellschaft).

**Conflicts of Interest:** VRVis Cockpits are used as the basis for basic research projects in applied industrial research at the VRVis Zentrum für Virtual Reality und Visualisierung Forschungs-GmbH (VRVis). Together with the VRVis, VRVis Cockpits can be employed as a data analytics tool in collaborative joint research projects. As such, VRVis Cockpits stand as a prime example for an interactive visual analytics solutions for time-oriented data. VRVis Cockpits are an interactive tool for the interactive analysis of large amounts of data, which offers the possibility for providing customized applications. The software has been developed at the VRVis Zentrum für Virtual Reality

und Visualisierung Forschungs-GmbH over the last decade in research work together with renowned company partners. More than 25 scientific papers about the analytical possibilities of VRVis Cockpits have been published, and several of these publications received awards. To continue the successful application of visual analytics in industrial applications, VRVis Cockpits are further developed in a spin-off company. The Visplore GmbH founded in July 2020 is responsible for the distribution of the standard software packages. The authors further declare that they have no affiliation with or involvement in the company Visplore GmbH, nor any financial, business, or personal interests, such as honoraria, educational grants, consultancies, stock ownership, or patent-licensing arrangements, regarding the software discussed in this manuscript.

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
