# Peer review of "Visual Analytics for Climate Change Detection in Meteorological Time-Series"

_forecasting, doi:10.3390/forecast3020018_

Round 1

Reviewer 1 Report

The authors have provided the readers with an interesting demonstration about using visual analytics for time-series climate data. I think the paper is well organized and all results and figures are clearly presented. However, I am not sure if this VA platform/tool is available online as a open-source package? It seems that these visualization techniques are just the common ones. Why VA is much better compared to other open-source libraries like Echart (https://echarts.apache.org/examples/en/index.html) or D3JS?

Author Response

First, we would like to thank the Reviewer for the time and effort put into reviewing our manuscript.

We appreciate your comment and understand your concern. As there are indeed other VA tools and methods available, these are mostly commercial systems (e.g. Tableau, QlikView, MS Power BI) that do not interact closely with their users. The VA platform used for our analysis is a result of several years of research projects with our partners from the industry. All the features of developed system are attentively tailored, allowing research-based further development and customer-based customization of developed analytical ensembles (here, we refer to these as analytical cockpits). In short, the analytical cockpits offer the subsequent users of the technology the opportunity to participate directly in the development process to ensure that the developed solution can really be well integrated into the existing work process.

In contrast to other stand-alone analytical modules and data visualizations (e.g. D3JS) which require a substantial knowledge and expertise to combine them into functional/informative dashboards, the advantage of our system lies in offering its pre-assembled analytical cockpits, whose individual functionalities resulted from the multi-year collaborative effort described above. These cockpits offer an extremely responsive design, achieved through a multi-threaded architecture, allowing instantaneous feedback and smooth interaction with the data across diverse set of analytical modules. This means that, once a user selects a data point or data subset of interest in one analytical module, the corresponding data points are highlighted in another module. This allows for a smooth multidimensional visual data exploration.

Some of these aspects are mentioned in our previous work, which, in an effort to avoid repetition, is only provided as a reference in the current manuscript. We also did not intend to promote the tool itself, but rather to highlight the benefits of using Visual Analytics approach in climate research. However, we have revised the subsection 2.2 (and throughout the paper) to elaborate more on some functionalities of the VA system used that differentiate it from the other solutions mentioned.

However, as a research company, we strive to constantly develop and improve these analytical cockpits and analysis techniques of our VA system. As such, many of the findings of the last 10 years or so are currently being further developed in a web-based tool. As this is still work in progress, we feel that it is still early to reference this effort in our current manuscript. However, we indeed do plan to release parts of our solution as Open Source application once it is fully shaped.

Reviewer 2 Report

This work is suitable having strong novelty , results and conclusion . 

  1. for long and short time span 45 years and 5 years are selected. Any reason behind this particular values?
  2. Improve the Figure 1,2,3,4 its impossible to read.
  3. from the heat map it is clear that earth temperature is increasing. However, can this data set tells why or what is the main reason or list the key factors which has contribution to increase this heat map?
  4.  this VA analysis tells how the temperature is rising but it would be good to shed some lights on the factors which influence them

Author Response

First, we would like to thank the Reviewer for the time and effort put into reviewing our manuscript.

We appreciate your comments and we have tried to address all the points raised to the best of our abilities.

Q1: The selection of this particular long-term time span was driven by the fact that, when focusing on climate investigations, the data records of at least 30 years are considered appropriate to fully capture the variability of climate conditions and their tendencies. The selection of the short-term time span of 5 years was in part driven by the initial findings derived from the long-term records, stressing that the last 7 years are showing a notable and consistent warming tendencies. This was further reduced to 5 years due to the data availability stemming from selected weather stations. In an effort to bring more clarity of this issue in our manuscript, we have elaborated more on these aspects. Please refer to the subsection 1.3.

Q2: This has been addressed.

Q3: Given that the selected stations are located within densely built urban areas, a part of this warming may indeed be attributed to climate change. However, a part may also be attributed to the degree of local urbanization and related local effects. To limit the local effect in our analysis, we have focused on those case studies that are similar to a certain extent, e.g. high-density cities with intensive traffic levels and being of the same Köppen climate classification, thus having more controlled background climate conditions and respective urban influences. However, we have tried to be more specific about these aspects. Please refer to the opening paragraph of Section 2 and subsection 3.1.

Q4: Indeed, we fully agree with this observation. It would be highly beneficial to highlight which aspects (e.g. traffic levels, degree of urbanization, population) affect the changes in local climate the most. As this requires additional dataset and additional comparative analysis, we are currently pursuing such a thorough investigation as our future research direction. As such, we have reflected on this in the newly added section on future research directions (please see section 5).

Reviewer 3 Report

The paper “Visual Analytics for Climate Change Detection in Meteorological Time-Series” is an application paper describing the analysis of a set of meteorological time-series using a previously developed VA tool. 

The paper investigates the potential of VA to detect and analyze evidence of climate change from time series of meteorological parameters generated by city based weather stations from three central European cities.

This is done using a VA tool composed of coordinated views of standard visual analytics techniques which are previously published. So there is no novelty from a VA point of view, but the contribution is rather on the analysis that it makes possible.

Overall, the paper is well written and reads easily. It offers an exhaustive introduction about the topic and a good motivation for the work. The structure of the paper is good and the details in the data analysis descriptions sufficient.

There are some clarifications and revisions needed in the paper which are outlined below. 

Although a previously developed VA system is used and referenced for the analysis, I would have appreciated a more detailed explanation/description of the views used for the analysis. What are the highlighted data points in the time-series distribution views of figures 1,2 and 5 for example? 

In figures 1,2 and 4 the histogram view compares the timespan 2016-2020 with the rest of the time period. The authors note that they make this distinction “Due to a number of initial observations described below”. What these are and why this time period was chosen is unclear. Do these observations relate to the short term data? Please clarify.

Also, the authors note that “... from the frequency distribution graph (Figure 1, down right) we can see that the temperatures exceeding 35 °C are more prominent in the last 5 years and that all of such instances are clustered in the last 15 years.” The 5 year observations are made based on the preset timespan. But where can these clustered instances of the last 15 years be seen in the histogram? Please explain. The same applies to figure 2 and the comment about the low temperatures not occurring in the last 2 years.

In the introduction and in the conclusions the authors mention that VA can handle massive and dynamic data streams, thus allowing for discovery of new relevant insights and patterns. Although this may be true in the system used by the authors, there are no examples of streaming data analysis in the paper and the data used in the study is not massive. Please either rephrase or provide additional information. 

The analysis presented in the paper is based on measurements from a single station for each of the cities. Some discussion is needed concerning the potential bias of using such single station measurements for detecting climate change. 

In addition, a suggestion for compensating for such bias,  for example, would be to compare the measured parameters with well established model data.

The authors use stacked time-series distributions of the min/max/avg parameter values for showing for example increase of temperature values over time. The seasonal character of the data however disrupts the continuity of the observation. It would be interesting to aggregate the time series by natural variability summer/winter (i.e. show only the summer respective winter values) or day/night to better highlight the parameter variation.

Some discussion on the scalability of the system and the analysis it makes possible is needed. How does the system perform in case of more time series, what is the optimal number of time series that can be visually analyzed in the proposed approach.

There is an overlap between this study and the previous published work [21], the authors should more clearly describe the contribution of the current work compared with the previous one using the same methods / VA tool.

An expert evaluation to confirm the presented analysis findings would be needed.

Author Response

The paper “Visual Analytics for Climate Change Detection in Meteorological Time-Series” is an application paper describing the analysis of a set of meteorological time-series using a previously developed VA tool. The paper investigates the potential of VA to detect and analyze evidence of climate change from time series of meteorological parameters generated by city based weather stations from three central European cities. This is done using a VA tool composed of coordinated views of standard visual analytics techniques which are previously published. So there is no novelty from a VA point of view, but the contribution is rather on the analysis that it makes possible. Overall, the paper is well written and reads easily. It offers an exhaustive introduction about the topic and a good motivation for the work. The structure of the paper is good and the details in the data analysis descriptions sufficient.

There are some clarifications and revisions needed in the paper which are outlined below. 

Although a previously developed VA system is used and referenced for the analysis, I would have appreciated a more detailed explanation/description of the views used for the analysis. What are the highlighted data points in the time-series distribution views of figures 1,2 and 5 for example? 

Answer: First, we would like to thank the Reviewer for the time and effort put into reviewing our manuscript.

We appreciate you comment and recognize the benefit of providing more information on the used VA system and its functionalities. As these are explained in detail in our previous work, and in a further effort to avoid repetition, we only provided a reference to our previous work in the current manuscript. However, we now see that some degree of additional explanation is needed.

The used VA system offers a number of pre-assembled analytical cockpits that offer an extremely responsive design, achieved through a multi-threaded architecture, allowing instantaneous feedback and smooth interaction with the data across diverse set of analytical modules. This allows for a smooth multidimensional visual data exploration. This means that, once a user selects a data point or data subset of interest in one analytical module, the corresponding data points are highlighted in another module. As such, in these particular figures (especially 1 and 2), we selected the target ranges from the histogram view which are immediately highlighted in the time-series view. In this way we could visually investigate which time periods are representative of those conditions where e.g. air temperature exceeds 35 degrees Celsius. To address this comment, we have added some additional explanation in subsection 2.2, and further elaborated on views used for the analysis (see subsection 3.1).

In figures 1,2 and 4 the histogram view compares the timespan 2016-2020 with the rest of the time period. The authors note that they make this distinction “Due to a number of initial observations described below”. What these are and why this time period was chosen is unclear. Do these observations relate to the short term data? Please clarify.

Answer: Thank you for pointing this out. Our initial long-term observations revealed a warming trend occurring in the last 7 years, starting from 2014, but being more prominent in last 5 years. This additionally aligns with our target short-term period, which relates to the period from 2016 to 2020. We thus aimed to further highlight this occurrence in our long-term analysis as well, by sub-setting the 2016-2020 data and making a comparative analysis with the rest. In referred figures, the heatmap view shows the progression throughout the years, whereby the Time-Series and Histogram views further offer a comparative analysis of the 2016-2020 subset versus the rest (1975-2016), which further highlights the abrupt changes in the recent years when compared to the 30 years past. We have reflected on this aspect in our revised manuscript (please see subsection 3.1).

Also, the authors note that “... from the frequency distribution graph (Figure 1, down right) we can see that the temperatures exceeding 35 °C are more prominent in the last 5 years and that all of such instances are clustered in the last 15 years.” The 5 year observations are made based on the preset timespan. But where can these clustered instances of the last 15 years be seen in the histogram? Please explain. The same applies to figure 2 and the comment about the low temperatures not occurring in the last 2 years.

Answer: We understand your concern. The answer to this point is closely connected to our first response above, which further relates to the core functionalities of the VA system used. Due to its inherent interactivity and multi-threaded architecture (e.g. the ability of achieving multiple executions of an action concurrently), our system is highly responsive and allows instantaneous visual feedback of a certain action (e.g. selection of a data point or a data subset) across all views (modules) in a single analytical cockpit (e.g. dashboard). In summary, if we select a particular value from a histogram view, all such instances will be immediately highlighted in other view – meaning that, for e.g. temperatures of and above 35 °C, the highlighted points in the Time-Series view reveal when exactly did such conditions occur. We hope that, by addressing the comment above in our revised manuscript, these non-clarities will be remedied for other aspects in our manuscript as well.

In the introduction and in the conclusions the authors mention that VA can handle massive and dynamic data streams, thus allowing for discovery of new relevant insights and patterns. Although this may be true in the system used by the authors, there are no examples of streaming data analysis in the paper and the data used in the study is not massive. Please either rephrase or provide additional information. 

Answer: We have rephrased the potentially misleading statements as per your suggestions. We decided to revert to the term ‘multifaceted’ instead.

The analysis presented in the paper is based on measurements from a single station for each of the cities. Some discussion is needed concerning the potential bias of using such single station measurements for detecting climate change. In addition, a suggestion for compensating for such bias, for example, would be to compare the measured parameters with well established model data.

Answer: We understand your concern. In an effort to raise attention to such limiting aspects of our present study, we introduced a new section in our revised manuscript (please refer to section 5), where we discussed such limitations and introduced some future research directions that will address these issues.

The authors use stacked time-series distributions of the min/max/avg parameter values for showing for example increase of temperature values over time. The seasonal character of the data however disrupts the continuity of the observation. It would be interesting to aggregate the time series by natural variability summer/winter (i.e. show only the summer respective winter values) or day/night to better highlight the parameter variation.

Answer: We agree with your comment. This particular aggregation was driven by the input data used for the long-term analysis. This was also one of the core objectives of our paper, where we intended to stress the importance of data resolution, data aggregation, and the amount of insights one can derive given the varying temporal scale of the underlying data. The long-term data was provided in monthly min/max/avg parameter values, whereby the short-term data came in an hourly format. As such, the seasonality was discussed in connection to the short-term data, as we could be more precise on such aspects here. We more strongly focused on seasonality and day-night differentiation in our previous contribution (referenced in this current manuscript). However, as we here aimed to derive insights on a higher level, we still discussed implications for seasonal changes and more frequent occurrence of e.g. tropical nights.

Some discussion on the scalability of the system and the analysis it makes possible is needed. How does the system perform in case of more time series, what is the optimal number of time series that can be visually analyzed in the proposed approach.

Answer: The used VA system is built to handle large amounts data. In our daily work, we handle more than 100 time-series at a time, which are simultaneously considered and comparatively analyzed. In our current manuscript, we also wanted to focus on open data sources (e.g. freely available) and what is commonly available to any other researcher, so as to achieve transferability and accessibility of our work to others for validation purposes, but also to encourage further development and take-up of our efforts.   

There is an overlap between this study and the previous published work [21], the authors should more clearly describe the contribution of the current work compared with the previous one using the same methods / VA tool.

Answer: We understand your concern. The previous contribution indeed drove the attention of using a VA approach in meteorological time-series analysis. However, this previous work is essentially different, as it did not focus on climate change, rather is was a comparison between different meteorological datasets – one depicting the real ground-level measurements, and the other, depicting a synthetically generated typical meteorological year (TMY). As such, the study concerned only the comparative analysis of one year worth of data. Nevertheless, we reflected on this in our revised manuscript (please refer to the section 4).

An expert evaluation to confirm the presented analysis findings would be needed.

Answer: We agree that this is an essential aspect to consider. The first author has an extensive background in the field of environmental studies and climate research, both empirical and numerical, and a proven track record on this part. As mentioned throughout the manuscript, we aimed to demonstrate the application potential of a VA approach in the context of climate research, which resulted from the ongoing research efforts from the said author.